# Climate Change and Its Influence on the Active Layer Depth in Central Yakutia

**Alexey Desyatkin** [1,2,*], **Pavel Fedorov** [1], **Nikolay Filippov** [1] and **Roman Desyatkin** [1]

[1] Institute for Biological Problems of Cryolithozone, Siberian Branch, Russian Academy of Science, Yakutsk 677980, Russia; baibal@yandex.ru (P.F.); finiva88@mail.ru (N.F.); rvdes@ibpc.ysn.ru (R.D.)

[2] Melnikov Permafrost Institute, Siberian branch, Russian Academy of Science, Yakutsk 677010, Russia

\* Correspondence: desyatkinar@rambler.ru; Tel.: +7-964-420-8307

**Abstract:** Analysis of climatic conditions for the period of instrumental measurement in Central Yakutia showed three periods with two different mean annual air temperature (MAAT) shifts. These periods were divided into 1930–1987 (base period A), 1988–2006 (period B) and 2007–2018 (period C) timelines. The MAAT during these three periods amounted −10.3, −8.6 and −7.4 °C, respectively. Measurement of active layer depth (ALD) of permafrost pale soil under the forest (natural) and arable land (anthropogenic) were carried out during 1990–2018 period. MAAT change for this period affected an early transition of negative temperatures to positive and a later establishment of negative temperatures. Additionally, a shortening of the winter season and an extension of the duration of days with positive temperatures was found. Since the permafrost has a significant impact on soil moisture and thermal regimes, the deepening of ALD plays a negative role for studied soils. An increase in the ALD can cause thawing of underground ice and lead to degradation of the ice-rich permafrost. This thaw process causes a change of the ecological balance and leads to the destruction of natural landscapes, sometimes with a complete or prolonged loss of their biological productivity. During this observation (1990–2018 period) the active layer of permafrost is characterized by high dynamics, depending on climatic parameters such as air temperature, as well as thickness and duration of snow cover. A significant increase in ALD of forest permafrost soils—by 80 cm and 65 cm—on arable land was measured during the observation period (28 years).

**Keywords:** permafrost; forest soils; climate change; meteorological conditions; soil thermal regime; active layer depth

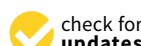



## 1. Introduction

The highest rate of warming is observed in the northern latitudes of the Earth [1]. Central Yakutia is located in this risky zone and characterized by the presence of underground ice wedges—an ice complex. Fluctuations in climate conditions during the early Holocene led to extensive thermokarst formation caused by the degradation (i.e., thawing) of the ice-rich permafrost deposits and subsequent surface subsidence [2]. However, on the Central Yakutia lowland, about 50% of ice wedges remain in upper permafrost and can thaw [3]. Under present warming, on the areas with a shallow ice complex, an increase in active layer depth (ALD) leads to the thawing of ice and further degradation of landscapes surface. This phenomenon plays a negative role in human life and agriculture. Thus, over the last decades, the destruction of constructions, the abandonment of arable land and other negative consequences of an increase in ALD have already been observed in this territory.

The ALD is an integral indicator of the heat supply of permafrost soils in the region of perennially frozen grounds distribution. It plays a leading role in the development of biotic indicators of soil formation processes, such as the root system of plants, the vital activity of soil microorganisms and the intensity of biological and biogeochemical transformations. However, the most important role of the ALD in conditions of close permafrost occurrence

is protecting the upper part of permafrost from thawing. Taking into account the delicate equilibrium of permafrost, we can assume the presence of changes in the dynamics of the ALD under changing climatic parameters. First of all, the changing of MAAT can play a primary role in the fluctuation of ALD. The secondary parameter is the fluctuation of precipitation, especially thickness and duration of snow cover. In recent decades, the problem of climate change on the planet has become an extremely topical issue. According to the World Meteorological Organization, the average temperature on the Earth has now increased by 0.85 °C. In Russia, this is equal to 1.29 °C, in comparison with the second half of the 19th century [4]. Further, according to IPCC forecasts, if the air temperature on the Earth's surface increases by 1.5 °C, then this may lead to the disappearance of 13% of existing terrestrial ecosystems [1,5,6]. The change in annual precipitation is also observed together with temperature increase [1]. The tendency towards the increase of annual precipitation (7.2 mm/10 years), mostly due to the increase of spring rainfall is observed over the territory of Russia. The decrease in winter precipitation is observed in the north-east of Siberia [1]. Snow cover is one of the primary environmental components in cold areas. Depending on thickness, snow, being a bad heat carrier, prevents heat exchange between soil and air and significantly protecting soil from intense cooling in winter. Time of beginning, duration, depth and loss of snow cover has a significant effect on the soil temperature regime and consequently on ALD [7–10]. Thus, snow cover prevents even seasonal soil freezing in the territories, where MAAT is close to 0 °C [7]. However, during the last decades the reduction of the area and the decrease of snow cover depth on the Earth is observed. That is, giving a positive effect for saving the upper permafrost until the present time. In connection with the ongoing changes on the planet, there has been an increase in the number of works indicating global climate changes and their impact on the regional climate of northern latitudes and on the parameters of the permafrost region [11–18]. Analysis of meteorological data during 1966–2016 period identified increasing MAAT trend in Yakutsk by 3 °C, in Vilyuisk by 2.2 °C, in Verkhoyansk by 1.8 °C and Oymyakon by 2.2 °C during the last 50 years [19–23]. An increase in the MAAT by 2–3 °C over the past decades has caused an increase in the temperature of the upper part of permafrost by 0.4–1.3 °C, which is associated with an increase in ALD and causes the activation of cryogenic processes [24–31]. Since the Central Yakutia is characterized by the presence of ice complex, the increasing ALD will lead to a strong degradation of the existing landscapes [32].

The long-term dynamics of the ALD in Central Yakutia is still not well studied. In literature, the dynamics of ALD and its forecast in the Arctic territories are more or less detailed [33,34]. They show an increase in ALD mostly in coastal and tundra landscapes without strong anthropogenic influence. This work reflects the results of long-term observations (1990–2018) on the dynamics of ALD in pale soils in natural, developing under the canopy of larch taiga, and in disturbed soil of arable land created as a result of uprooting the larch forest. The aim of this work is to characterize the dynamics of soils ALD under the natural and anthropogenic condition, during the long-term period depending on the dynamics of climatic factors. For this purpose, a study of the peculiarities of climate dynamics, ALD dynamics in Central Yakutia and analysis of the peculiarities of ALD changes from climatic characteristics was fulfilled.

## 2. Materials and Methods

### 2.1. Study Area

The study of ALD dynamics during the 1990–2018 period was carried out on the territory of the Lena–Amga interfluve at 40 km east of Yakutsk around Tyungyulyu settlement in the model alas of the Institute for Biological Problems of Cryolithozone SB RAS (Figure 1).

The northern part of the Lena-Amga interfluve can be characterized as an alluvial terraced plain., The main terraces on this area are Bestyakh (140–163 m above sea level), Tyungyulyu (150–183 m) and Abalakh (201–219 m) terraces. Alas relief formed by thermokarst depressions is developed on the last two, which shows a strong distribu-

tion of underground ice. On the Tyungyulyu terrace, the river valleys are very poorly developed—there is an evenness and weak dissection of the relief. The relief is complicated by ancient erosional depressions that continue similar depressions of the Bestyakh terrace. Here non-layered ancient sands are covered with sandy loams and loams, representing their floodplain facies, with a total thickness of about 6–10 m. This stratum includes ice wedges and represents an ice complex formed during the Pleistocene glaciation. The depth of the upper surface of the ice complex on the Tyungyulyu terrace often exceeds 3 and more meters [35]. The depth of ice wedges reaches 40 and more meters, the volume of ice occupies up to 40–50% of the ice complex. During the thawing of the ice complex in the Holocene, there was a massive formation of alas depressions, which occupy up to 20–30% of the entire territory [2]. The permafrost temperature changes depending on the landscape and geological situations. On the vast inter-alas area, covered by larch forest, the average annual temperature of permafrost is −2 to −6 °C, at the same time, on treeless areas in alas depressions, as well as in dry areas composed by sandy soils, it is not lower than −1 °C [3]. The climate of the research area is sharply continental with long, cold winters (up to 7 months) and short but hot summers (3 months). Around 185 to 340 mm of precipitation falls during a year with evaporation over the summer up to 400 mm [36]. The first "forest" site is located under the larch forest, the second site is on the arable land created during the uprooting of larch-cowberry forest in 1985–86 (Figure 1).

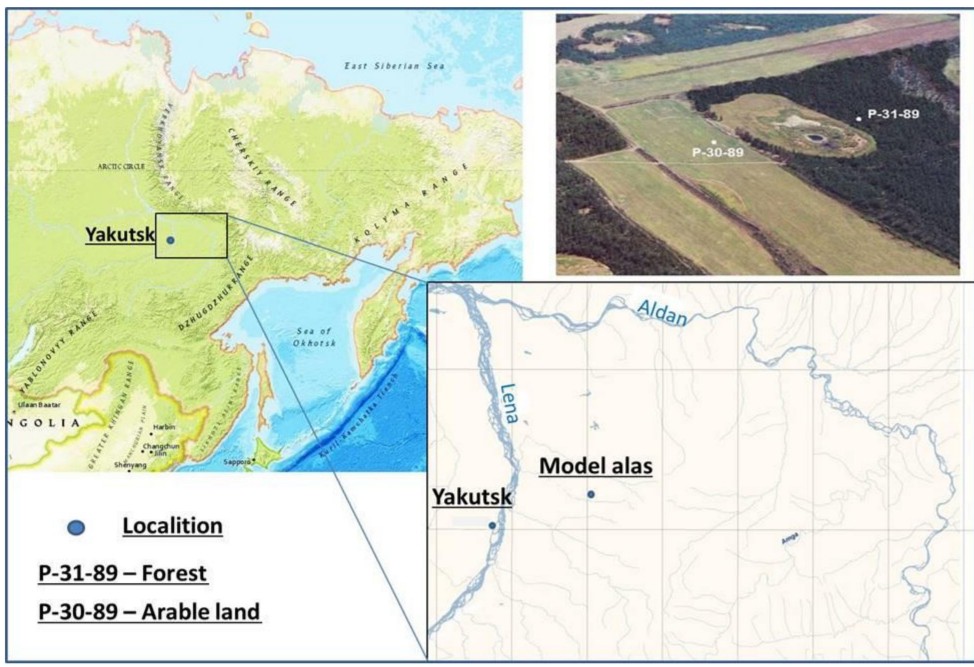

**Figure 1.** Study area and location of forest (right) and arable land (left) plots.

*2.2. Materials*

The air temperature data were obtained from the Roshydromet (RIHMI-WDC) database for Yakutsk station (www.meteo.ru/climate/sp_clim.php).

The properties of two sites for ALD study were followed; Site 1 (P-31-89) is located south of model alas in larch forest (*Larix Cajanderi*) with cowberry (*Vaccinium vitis-idaea L.*) (Figure 1). Forest crown density is 0.5–0.6. The height of the trees is 11–12 m, with a diameter of up to 20–25 cm. The forest has the second layer which is composed of young larch of 5–6 m high. Ground vegetation is represented by cowberry with little involvement of sedge (Carex capillaris), crowberry (*Empetrum nigrum*), peavine (*Lathyrus humilis (Ser.) Spreng*), and pyrola (*Pyrola asarifolia Michaux*), the projective coverage is 70%. The relief of the surface is hillocky fissured. The height of the hillocks is up to 7–10 cm; the diameter is 70–100 cm. The visible width of the fissures filled with litter is up to 20–30 cm. Soil profile P-31-89

was studied on September 20 of 1989. The morphological structure of soil was as follows: O1 0–2(4) cm (litter); AY (10YR2/2) 2(4) –9 cm; Aye (10YR4/2) 9–15 cm; BPLCA (10YR4/3) 15–41 cm; BCA1 (10YR5/3) 41–68 cm; BCA2 (10YR5/4) 68–88 cm; DCA (10YR7/4) 88–138 cm. According to the "Classification and Diagnostics of Russia Soils" [37], the soil belongs to the soddy-pale-solodized (Haplic Cryosol - WRB), the traditional name is permafrost pale solodic with the morphological structure: AO - A - A1A2 - B2 - B1Ca - B2Ca - D [38,39].

Site 2 (P-30-89) was located north of the model alas on the arable land (Figure 1). The relief is flat, with no vegetation. Soil profile P-30-89 was studied at the same time with a forest plot. Morphological structure of the soil is following: P (10YR5/2) 0–15 cm; BPLad (10YR6/2) 15–25 cm; BPLCA (10YR4/4) 25–35 cm; BCA (10YR5/4) 35–71 cm; B (10YR6/3) 71–96 cm; D (10YR7/4) 96–200 cm. This soil according to "Classification and Diagnostics of Russia Soils" [37] refers to light solonetzic agrozems (Calcic Mollic Solonetz - WRB).

Granulometric composition of the studied permafrost pale soils has a binomial structure. The upper part of the soil under the forest is represented by light fraction and on cultivated soil by a medium loamy fraction. A slight increase in the silt fraction in the BPLCA horizon of the soil under the forest is apparently due to the weak removal of silt particles from the humus-accumulative and eluvial horizons to the underlying horizon. Granulometric composition of the lower part of profiles (from a depth of 35–43 cm) is represented by sandy loam and the lowest horizons by sand.

## 2.3. Methods

Time intervals obtained from the RIHMI-WDC data were selected based on an analysis of long-term variations in mean annual air temperature over the period 1930–2018 (Figure 2). Direct in-situ measurements of snow depth were carried out during the period 1988–2010. The difference in fluctuations in the snow depth of forest and arable land sites during the 23 years of direct measurement reached a maximum of 11–13 cm. In other years the difference in snow depth of two studied sites was only ± 3–4 cm. Taking into account the small difference in snow cover depth on the studied locations for a general characteristic of this indicator in a long-term series, we used the average values of two points. Moreover, these average indicators in 81% of cases coincide with the data of Roshydromet at the Yakutsk station (90% confidence level). Thus, to lengthen our data series for the last decade, we used the data of the Yakutsk weather station.

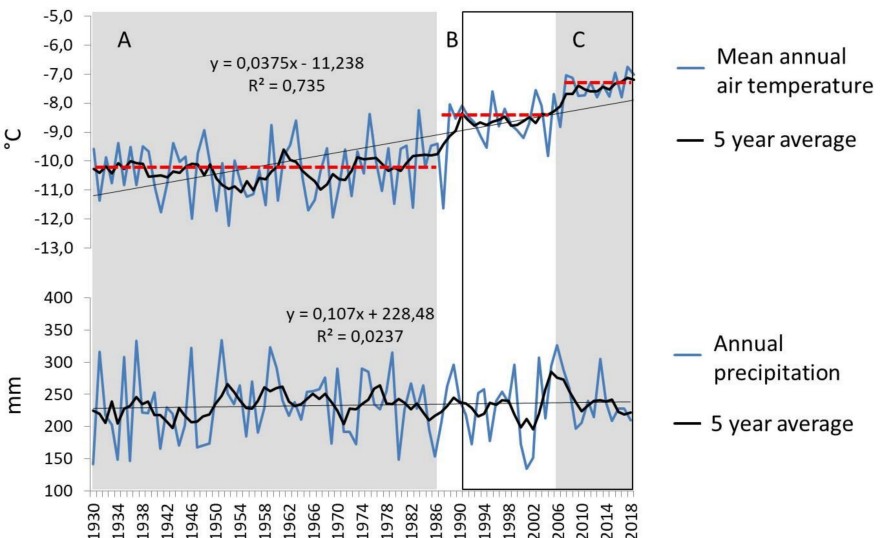

**Figure 2.** Climatic data for the 1930–2018 period (weather station Yakutsk). Red dotted line: periods of Table 1. In general, there is an increase in temperature for all months. The highest positive increase is noted in December and January during winter. In springtime in March, April and May and autumn season in October (Table 1). Precipitations showed a decrease in −4.7 mm during summer of C period.

**Table 1.** Characteristics of climate change during different periods of mean annual air temperature (MAAT).

| | Air temperature, C° | | | | | | Difference between periods |
|---|---|---|---|---|---|---|---|
| | Period B | | | Period C | | | |
| | min | max | average | min | max | average | |
| Jan | −45.0 | −32.0 | −38.1 | −41.4 | −33.8 | −36.2 | 1.9 |
| Feb | −37.5 | −27.4 | −33.3 | −37.2 | −27.7 | −33.2 | 0.1 |
| Mar | −25.3 | −12.0 | −20.3 | −22.3 | −10.8 | −17.3 | 3.0 |
| Apr | −9.4 | −0.5 | −4.7 | −5.8 | −0.1 | −2.8 | 1.8 |
| May | 5.0 | 9.8 | 7.4 | 6.6 | 10.2 | 8.9 | 1.5 |
| Jun | 13.7 | 20.2 | 16.5 | 15.5 | 19.5 | 17.5 | 1.0 |
| Jul | 17.1 | 23.0 | 19.6 | 16.7 | 22.5 | 20.0 | 0.4 |
| Aug | 13.2 | 18.2 | 15.3 | 13.4 | 17.6 | 16.2 | 0.8 |
| Sep | 3.5 | 9.4 | 6.1 | 3.8 | 8.2 | 6.3 | 0.2 |
| Oct | −11.1 | −3.8 | −7.5 | −8.2 | −3.7 | −6.1 | 1.4 |
| Nov | −32.0 | −20.3 | −26.5 | −29.0 | −22.7 | −26.1 | 0.4 |
| Dec | −43.8 | −33.8 | −37.6 | −40.0 | −30.2 | −35.8 | 1.8 |
| Year | −9.8 | −7.6 | −8.6 | −7.8 | −6.7 | −7.4 | 1.2 |
| Summer | 11.9 | 14.1 | 13.0 | 12.4 | 14.5 | 13.8 | 0.8 |
| Winter | −22.8 | −20.5 | −21.6 | −22.3 | −20.1 | −21.1 | 0.4 |
| | Precipitations, mm | | | | | | |
| | Period B | | | Period C | | | Difference between periods |
| | min | max | average | min | max | average | |
| Jan | 3.4 | 15.8 | 8.4 | 4.7 | 21.9 | 10.5 | 2.1 |
| Feb | 2.8 | 22.2 | 8.3 | 1.4 | 15.2 | 7.3 | −1.0 |
| Mar | 0.5 | 13.7 | 6.2 | 1.2 | 20.7 | 6.4 | 0.2 |
| Apr | 0.3 | 15.6 | 5.9 | 1.4 | 17.9 | 10.0 | 4.2 |
| May | 1.1 | 57.8 | 19.7 | 3.8 | 49.9 | 20.5 | 0.9 |
| Jun | 5.5 | 88.6 | 30.5 | 3.1 | 60.4 | 28.8 | −1.7 |
| Jul | 4.0 | 100.1 | 43.0 | 15.0 | 74.6 | 41.9 | −1.1 |
| Aug | 2.5 | 150.8 | 35.7 | 9.2 | 71.9 | 37.4 | 1.7 |
| Sep | 3.9 | 86.4 | 32.6 | 8.6 | 67.2 | 28.2 | −4.5 |
| Oct | 4.0 | 38.3 | 17.8 | 6.1 | 35.3 | 19.1 | 1.3 |
| Nov | 4.1 | 41.6 | 17.5 | 6.1 | 33.1 | 15.3 | −2.2 |
| Dec | 3.7 | 20.4 | 9.4 | 2.9 | 13.9 | 8.3 | −1.1 |
| Year | 147.7 | 326.2 | 234.9 | 196.9 | 304.9 | 231.8 | −3.1 |
| Summer | 81.9 | 255.6 | 161.5 | 121.9 | 240.3 | 156.7 | −4.7 |
| Winter | 42.0 | 100.9 | 73.4 | 56.3 | 97.1 | 75.0 | 1.6 |

ALD in warm-season was measured by soil auger, to the permafrost table along the observation period. To analyze the dynamics of ALD in a long-term regime, we used the maximum depth for the season, which, depending on the meteorological conditions corresponds to the last decade of August or the end of the second decade of September. ALD measurements using the auger method is the most accurate method with an error of about 1 cm. According to the Russian Standards of Hydrometeorology, the temperature sensors at a depth of more than 1 meter are located with a large interval. Standard temperature measurement depths are 0.2, 0.4, 0.6, 0.8, 1.0, 1.2, 1.6, 2.4 and 3.2 m. Thus, in our case, with thawing depth 3 m, when measuring temperatures at standard depths, we would not know the real ALD within the interval between 2.4–3.2 m. In the early years of research, measurements were made in triplicate. The distance between measurement of adjacent dates was at least 2 m. After analyzing these data in subsequent years, when data homogeneity was revealed, the replication was reduced to 1 time per period. It should be taken into account that during the autumn freezing and summer thawing, the soils of the active layer are deformed and lead to the healing of previously drilled boreholes.

A statistical approach of Kolmogorov–Smirnov Test (K-S test) and t-test were used in the analysis of obtained data.

## 3. Results

### 3.1. Climate Changes

The analysis of the Roshydromet data during the period of instrumental observations (1930–2018) showed two shifts of the MAAT (warming) at Yakutsk station. From 1930 to 1987, a stable period with small fluctuations was observed, with MAAT −10.3 °C (base period A). Then occurred the first shift of MAAT, and from 1988 to 2006 there comes a period B with a sharp increase in the MAAT by 1.7 °C (t-value = −6.4, p < 0.001). The third step was observing the last decades after the second shift of MAAT in 2007, and lasting till 2018, where the MAAT for this period is amounted −7.4 °C (Table 1). A significant statistical difference between these two MAAT shifts (t-value = −4.7, p < 0.001) was revealed. Thus, over the past 88 years in Central Yakutia, there has been an increase in the MAAT by 2.9 °C (Figure 2). The Kolmogorov–Smirnov Test (K-S test) of Normality showed a *p*-value 0.03010, which provides good evidence that MAAT data are not normally distributed.

According to the annual precipitation, there is a slightly positive trend in C period due to the increase in April and October precipitation. For the studied period from 1990 to 2018 the long-term value of precipitation is 235.6 mm, the amount of summer precipitation for the same period is 161.2 mm and accounts for 68% of the total precipitation. Precipitation amount between periods B and C have no significant difference, t-value = 0.4 and t-value = 0.3, respectively, with p < 0.05. The value of the K-S test statistic (D) is 0.06063 with the *p*-value of 0.87916, which shows that precipitation data have good normality during the two periods of B and C.

### 3.2. ALD

Basing on ongoing climate changes, it was established that on the territory of Central Yakutia there was an increase in ALD, which sharply increased during the period B (Figure 3e) [40]. Thus, the observations revealed that during the period B, the ALD of forest plot was increased by 60 cm, the maximum depth was observed in 2006 and reached 180 cm, the average of ALD during this period was 141.7 cm (t-value = 1.4 at p < 0.05). The arable land ALD increased by 40 cm during that period, with a maximum value of 280 cm. The average ALD of the arable land plot during period B was amounted by 257.6 cm. Such dynamics of ALD was due to a sharp increase in the mean annual air temperature by 1.7 °C after the first shift of MAAT. However, the annual precipitation amount remained almost stable, however, there was a certain trend towards the increase in the amount of summer rainfall and at the same time, the decrease of winter precipitation amount.

During the C period, the ALD of soil under the forest significantly increased but remained relatively stable. The maximum depth during this period is stably kept at around 170–180 cm and the average value is 168.7 cm. T-test showed the significant statistical difference of ALD between two periods (B and C) after MAAT shift (t-value = 4.2 at p < 0.001). On the arable land, ALD trend shows an increase by 10 cm. The average ALD in this period is 279 cm and reached a maximum depth of 295 cm in 2013. In general, the ALD on arable land is kept at 280–290 cm depth. The arable land ALD also shows the significant statistical difference between two periods with t-value = 3.8 at p < 0.001. During the study period, increase in temperature difference level between periods B and C amounted to 1.2 °C versus 1.70 °C in comparison with the A–B periods.

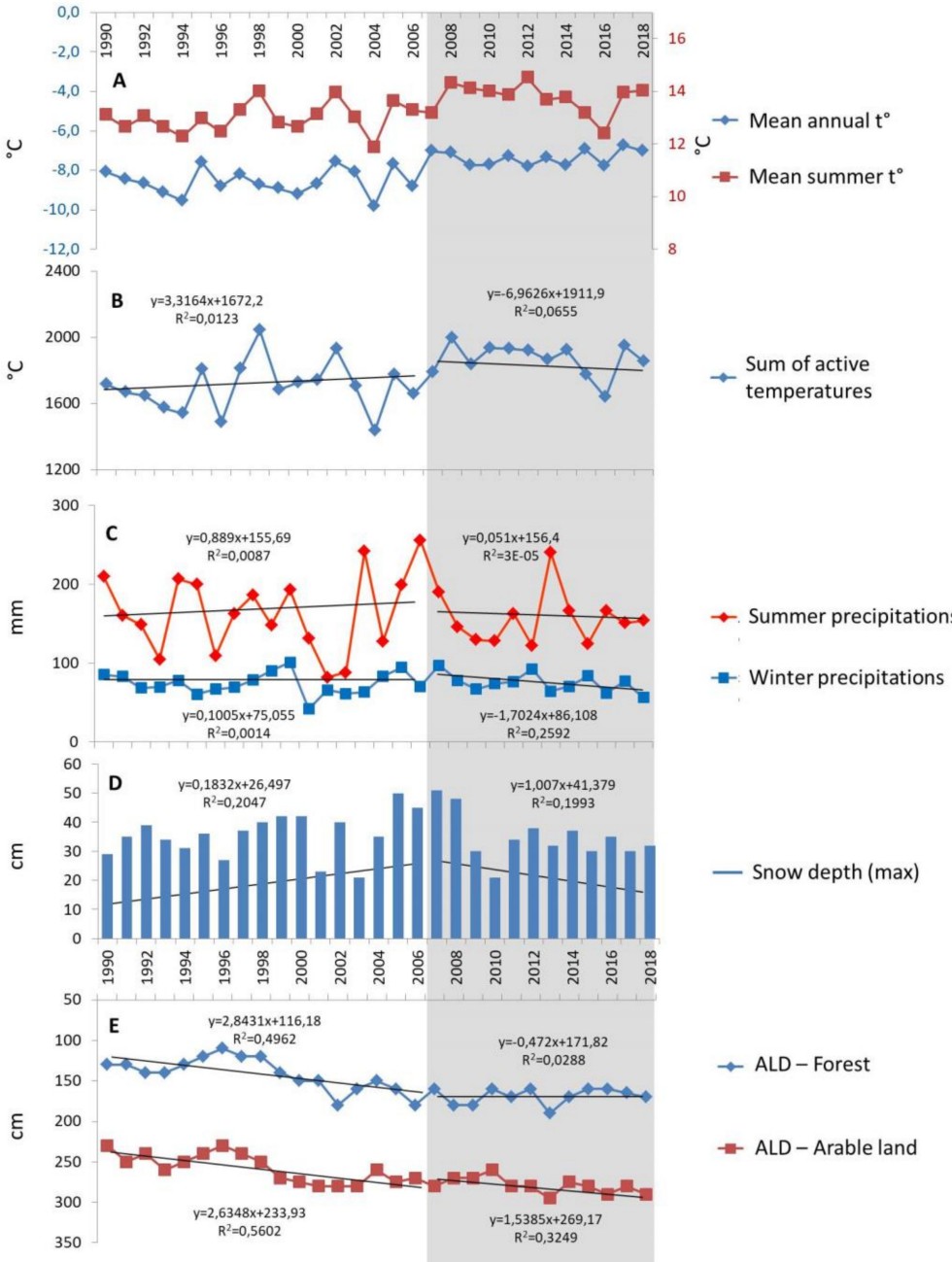

**Figure 3.** (**A**) The mean annual temperature***, (**B**) the sum of active temperatures higher than 10 °C***, (**C**) winter–summer precipitation*, (**D**) the maximum snow depth*, (**E**) the active layer depth (ALD) and their trends during the MAAT periods B and C***. White background: the second MAAT period B (1990–2006). The grey background is the Table 2007. Significance of difference between B and C period of MAAT; *-at p < 0.05, **-at p < 0.01, ***-at p < 0.001.

### 3.3. Climate Condition Influence on ALD

In the Russian permafrost zone, an increase in soil temperature by 0.8–1.6 °C has been observed over the past four decades at depths of 320 cm. There is a general long-term tendency for a climate-related increase ALD in the north of the Far East and Yakutia [41]. To identify the relationship of depth with various climatic parameters, a correlation analysis was performed (Table 2).

The time period of soil ALD measurement (1990–2018) is within the MAAT periods B and C. The highest summer temperatures were noted in 1991, 1995, 2010, 2011, 2014 and 2017, and during those years the maximum daily air temperatures reached 28 °C in July. Minimal air temperatures ($<-45$ °C) were observed during the whole period B, from 1988 to 2006, the daily air temperatures from $-46$ to $-51.1$ °C were stably recorded.

The period C started in 2007 was characterized by warmer winters. Minimal temperatures are observed during the winters from 2010 to 2014 when the minimum values of $-45.2$ to $-49.0$ °C were recorded. In subsequent years, the temperatures below $-45$ °C were not fixed. The revealed changes in climate variables indicate a decrease in the severity of the continental climate of the studied area.

According to data from the Yakutsk weather station for the period 1930–2018, in 80% of cases, there is an uneven distribution of precipitation over the summer months with the main amount in the second half of summer [36,41]. Thus, the noticeable influence of rainfall on the soils thawing is not so common in the region, taking into consideration the fact that summer seasons with heavy rain are quite rare [41]. During the study period, there is a wide variation in the amount of precipitation. At the period B, the annual precipitation ranges from 147.7 to 326.2 mm with min/max in 2001 and 2006, respectively. The precipitation value of B period was 234.9 mm, whereas during the period C there was a fluctuation from 196.9 mm in 2009 to 304.9 mm in 2013 with a value 231.8 mm for the C period (Table 1). This shows a slight decrease of precipitation for the last decades from the long-term annual norm of 233.1 mm (1930–2018). The amount of summer precipitation also has large amplitude from year to year, but almost completely repeats the amount of annual precipitation, because 68% falls in the summer. At the last MAAT period C, there is a decrease in the amount of winter precipitation which shows a steady trend (Figure 3).

It has been revealed that the highest correlation with the ALD has a positive temperature of the spring season. Climate warming in the spring shows an increase in the correlation dependence with the monthly temperatures of March, April, and May (Table 2). There is also an increase in correlation with the October temperature, which proves the warming of the autumn period (Table 2). The mean annual and summer precipitation does not show such a close relationship, although the snow cover depth has a rather high correlation.

**Table 2.** Correlation coefficients ($R_2$) and t-test results of ALD and climatic parameters. Difference between periods B and C.

| Climatic Parameters | Forest | Arable land |
|---|---|---|
| | $R_2$ | $R_2$ |
| Mean annual t *** | 0.522 | 0.473 |
| Mean summer t *** | 0.527 | 0.363 |
| March ** | 0.379 | 0.245 |
| April ** | 0.428 | 0.423 |
| May * | 0.403 | 0.227 |
| October *** | 0.287 | 0.267 |
| Mean winter t * | 0.101 | 0.222 |
| Active t sum *** | 0.416 | 0.348 |
| Mean annual precip. *** | 0.075 | 0.093 |
| Mean summer precip. * | 0.122 | 0.132 |
| Snow depth * | 0.285 | 0.300 |

Significance of difference between B and C periods of MAAT; *-at $p < 0.05$, **-at $p < 0.01$, ***-at $p < 0.001$.

Thus, it can be assumed that the soil ALD in the study area depends mainly on temperature parameters. The highest impact has been shown by positive temperatures of spring/autumn months which are consequently increases a sum of active temperatures. Figure 3 show that during period B an increase in temperature was accompanied by an increase in the snow depth, which caused deepening of ALD on the forest and arable

land both. The C period has a relative a decrease in the sum of positive temperatures and decrease in snow depth. During this period forest soil ALD ceased the deepening, and arable land ALD increased by only 10 cm. On the arable land it is due to a greater supply of solar energy on the open surface, which is the main reason for the increase in the ALD in the warm season. Thus, in these two different plots in the taiga landscapes of the considered region, the inter-annual variations in the sum of air temperatures and precipitation of the warm period are not significant reasons for the inter-annual variability of seasonal ALD.

## 4. Discussion

As is known, the depth of seasonal thawing depends on the mechanical composition of soils, their moisture content, the nature of the vegetation and soil cover. In previous literature the main factors determining the long-term variability of ALD was the sum of positive air temperatures and summer precipitation [42–44]. However, recent works on this issue show that long-term changes of ALD and the sums of summer air temperatures do not correlate sufficiently well [45,46]. Additionally, some researchers show that for the permafrost distribution area, the correlation coefficients between summer precipitation and seasonal ALD are not more than 0.3–0.4 [7,45]. However, during the periods B and C these coefficients were only 0.122 and 0.132, respectively, which is an indicator of low relation of precipitations with the soil ALD. Thereby, in this work, this is one of the important reasons for variability of soil temperature [32].

The second reason for the increase in ALD in the studied areas is the lengthening of the warm period due to spring-autumn warming. The average value of daily air temperature transition through 0 °C for the base period A is considered as April 30, at which the least date fluctuation is observed [47]. The variation of dates for the entire period was up to 20 days. An analysis of meteorological data for the period 1930–2018, taking into account the parameters of recent decades, shows that there is a shift in the transition to earlier dates due to the onset of positive air temperatures in the second decade of April. In general, there is a shift towards early terms by 5 days in the long-term trend of the daily temperature transition dates through 0 °C for the period 1930–2018, which generally coincides with previously published data [40]. During the time of our observation (1990–2018), a steady shift of the spring and autumn dates of the temperature transition through 0 °C occurred. On average, the onset of positive temperatures during the observation period (1990–2018) shifted 9 days earlier, from the third decade of April to the second decade (t-value = 2.1 at p < 0.01).

The autumn transition from positive air temperature to negative also has a shift. In a long-term trend of the 1930–2018 period, the transition of air temperature showed a shift to later dates, which averaged +6 days [47]. Moreover, the scatter of dates for this period has 9 days range of min/max (09/19/1998 and 09/10/2009). Within period B, the earliest average transition of air temperature through 0 °C occurred on September 19, and since 2007 the average date of transition shifted to September 22. The latest date of the autumn transition is shifted from October 6 to October 9 in periods B and C, respectively. Thus, the onset of autumn negative temperatures shifted by 3 days (t-value = 0.5 at p < 0.05). Boike et. al. during the 2000–2011 period showed a warming trend, with an average increase of about 0.12 °C/year. The average rate of warming during the April–May transition period was 0.17 °C/year and 0.19 °C/year in the September–October period, but ranged up to 0.49 °C/year during September–October [48]. These data are well corresponded to ours.

The mean duration of the warm season of the period B was 158.5 days, and 164.2 days for period C. As a result of temperatures transition through 0 °C, an early start of thawing and later freezing of the soil occurred, which leads to a longer action of positive temperatures causing an increase in the depth of the ALD. The same changes in the transition of air temperatures from negative to positive were noted in the works of other researchers in the studied region [49,50].

Taking into account the results of studies on the influence of snow cover and its parameters on the soils temperature regime, a decrease in the area and in the snow cover depth has been noted over the past decades [7,18,51,52]. In our study, we analyzed the influence of the snow cover on the ALD on the studied plots.

During the measurement period, snow cover has undergone significant fluctuations. The maximum snow depth in the studied area is established in February–early April, during which the observation period of this indicator had significant fluctuations over the years. In general, during period B, the snow cover depth tends to increase. During the C period, there is a trend of decreasing snow depth (Figure 3). At the same time, there is a tendency for the duration of the snow period. On average, the establishment of snow cover in the fall was 5 days later and the spring melting was earlier by 5 days compared to the B period. The years with the highest snow were at the junction of MAAT shift (2004–2008), with the snow depth reaching 45–51 cm. The average snow depths for the periods B and C were 34.2 and 34.8 cm, respectively, showing no significant changes despite a decrease in winter precipitation. During the MAAT period B, an increase of the snow cover depth was also observed in parallel, which significantly changed the thermal regime of the "soil–snow–atmosphere" system toward warming due to thermal insulation properties, thereby preventing freezing of the soil. In general, in Siberia, a change in snow depth determines a change in average annual soil temperature up to 50% of cases [53]. Observed during the period C differences in ALD trends in the forest and arable land sites, in our opinion depend on differences in the content of soil moisture of these soils. Similar results were noted in the works of other researchers [11,13,18,52]. In sum, we can say that an increase in winter temperature is compensated by a decrease in the heat-insulating and warming role of the snow cover.

Thus, recent works show that short-term meteorological events are increasingly weakening and delayed with depth [18,53]. Further, the gradual increase in ALD is in line with the long-term trend of the atmosphere, especially the lengthening of the warm period of the year. Additionally, it will be better to use data mining techniques on meteorological reanalysis to develop a coherent framework for the identification of extreme climate conditions relevant for ALD deepening and a decline of permafrost extend as it was used in Western Siberia. Several key types of events have been classified based on various combinations of temperature, precipitation and snow depth statistics. Then, the respective events have been identified in ERAInterim reanalysis and evaluated against in-situ observations in the West Siberia region. The evaluation proved that the developed algorithm could successfully detect relevant extreme climate conditions in meteorological reanalysis dataset. It also indicated possibilities to improve the algorithm by refining definitions of extreme events. Nevertheless, the method is applicable for all kinds of gridded climatological datasets that contain air temperature, precipitation and snow depth [54].

## 5. Conclusions

As a result of this work, we found two shifts with increasing MAAT from 1930 to 2018 in Central Yakutia. During these periods, an early transition from negative air temperatures to positive and a later establishment of negative temperatures are observed. There is a shortening of the winter season and an extension of the duration of days with positive air temperatures

During the observation (1990–2018) the active layer of permafrost was characterized by high dynamics, depending on climatic parameters such as thickness and duration of snow cover, as well as air temperatures. During the MAAT period B, there was a significant increase in soil ALD by 60 cm in the forest and by 40 cm in arable land. During the C period, we also observed a deepening of ALD but in less rate than period B. In general, a significant increase in ALD of forest permafrost soils was observed—by 80 cm and 65 cm on arable land during 28 years of observation. Correlation analysis of the relationship between the ALD of soils and climatic variables revealed a relatively high degree of temperature

parameters influence. The highest influence on ALD was exerted by an increase in spring month temperatures.

This work, taking into account local MAAT shifts, showed the significant ALD deepening during 28 years of measurement in Central Yakutia. Future warming of climate may cause irreversible changes in landscapes with ice complex, not only under human activity but in natural conditions too.

**Author Contributions:** Conceptualization: A.D. and R.D.; Methodology, R.D. and A.D.; Investigation, all authors; Writing-Original Draft Preparation: A.D., R.D.; Writing-Review & Editing: A.D., R.D. and P.F. All authors have read and agreed to the published version of the manuscript.

**Funding:** The article was prepared as part of the SB RAS project on the topic 0376-2019-0006; registration number AAAA-A19-119040990002-1 and grant RFBR 19-29-05151\19.

**Data Availability Statement:** Data available on request due to restrictions e. g. privacy or ethical. The data presented in this study are available on request from the corresponding author. The data are not publicly available due to [one coauthor writing paper using this and related data, after his publication it will be available].

**Conflicts of Interest:** The authors declare no conflict of interest.

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
