# Peer review of "Climate Change and Its Influence on the Active Layer Depth in Central Yakutia"

_land, doi:10.3390/land10010003_

Round 1

Reviewer 1 Report

I would like to commend the authors on putting together an nteresting paper, correlating change in ALD with temperature and precipitation records.  In general, the research was well presented and the dataset of measured ALD over an almost 40 year period is quite extensive and impressive. The paper is well written, however i do encourage you to review to catch minor typos and editorial corrections that remain to be made. 

I'll first address a few comments drawn from your conclusions and then there are a further comments show below.  Your overall conclusions of your paper seem to be fairly standard and expected - whereby the increase in temperature, most critically the number of thawing degree days seems to correlate with the increase in ALD. However there were a few comments made in the conclusions that require further details/explanation.

  • You indicate that snow cover is significant, was the snow measured on site at each location or just from the Yakutia station? Would the vegetation cover not alter the snow cover dramatically between your two locations? What were those observed differences?
  • The final sentence seems confusing. Are you implying that there will be an ever increasing rate of ALD? In fact, in most numerical simulations even with an RCP of 8.5 the ALD eventually stabilizes and it is the MAGT that then slowly warms over a much longer period of time until the permafrost has completely thawed. The irreversible changes you describe, have those already occurred with the change in ALD? How much deeper do you expect the ALD to go? You differentiate between human activity and natural conditions - what does the distinction refer to? The change in air temperatures? How your ALD will be impacted?

Other comments include:

  • You a few times refer to permafrost melting. This must be corrected to permafrost thaw
  • The air temperatures were recorded in Yakutia, yet your site was 40 km away. There can be significant differences in temperature at precipitation at a distance of 40 km. Was this addressed in any way?
  • The measured ALD - what is the degree of error using the auger method?
    How many ALD measurements were taken each time? Over the course of 30 years of data collection that is significant disturbance to the soil... what distance relative to previous years was the augering completed?
  • How much would local heterogeneity in the soil profile and presence of ice influence your ALD measurement?
  • In your discussion you highlight the influence soil moisture has in the ALD, did you record the moisture content of your samples? Or conduct any thermal property analyses on your cores?
  • What is the depth of permafrost at your location?
  • The introduction and conclusion discuss in the ice rich nature of this soil, yet there was no discussion on how this would/could influence the depth of ALD at the measured locations.

Author Response

Dear Reviewer please see attached file.

Reviewer 2 Report

The manuscript addresses the problem of climate change impact on active layer deepening in Central Yakutia. The research is not novel on topic/methods, but it is very important since it contributes to a better understanding of the permafrost degradation in a remote, but vulnerable part of the cryosphere. 

Below are given some specific comments to the paper.

  • more information on the study area would be useful for the readers (relief, geology, land cover, type of permafrost etc.). It would be also good to write a paragraph about permafrost characteristics in the region (thickness, temperature, active layer etc.). Are there any other boreholes nearby? What is the distance to the Yakutsk meteorological station?
  • what is the accuracy of the method used here for the estimation of the active layer? You should refer to this in the Methodology. Is there any possibility that frozen material to occur within the active layer and thus the thickness of AL to be greater?
  • the evolution of the AL was interpreted based on only 2 boreholes. Are there any other similar measurements in central Yakutia which are also in line with your findings? It might be risky to extrapolate your findings on a larger scale. A solution would be to compare your climatic data with ERA-Interim model (for example) and active layer with ESA CCI active layer thickness and to see whether the dependencies described by you are also available for the entire region. Otherwise, these findings have local importance in my opinion. Here are listed some studies which showed a good agreement between climate data operated by Roshydromet and ERA-Interim:

    Cheţan, M.-A.; Dornik, A.; Ardelean, F.; Georgievski, G.; Hagemann, S.; Romanovsky, V.E.; Onaca, A.; Drozdov, D. 35 Years of Vegetation and Lake Dynamics in the Pechora Catchment, Russian European Arctic. Remote Sens. 2020, 12, 1863.

    Dee, D.P.; Uppala, S.M.; Simmons, A.J.; Berrisford, P.; Poli, P.; Kobayashi, S.; Andrae, U.; Balmaseda, M.A.; Balsamo, G.; Bauer, P.; Bechtold, P. The ERA-Interim reanalysis: configuration and performance of the data assimilation system. Q. J. R. Meteorol. Soc 2011, 137, 553–597.
  • You might also look at the land surface temperature evolution from MODIS data. In a recent paper (Boike et al. 2016) `the land surface temperatures showed a consistent warming trend, with an average increase of about 0.12 °C/year` in Central Yakutia. In the same paper snow cover is also assessed in this region, therefore it would be good to refer to this in the Discussion because several findings are similar to yours.
  • Boike, J.; Grau, T.; Heim, B.; Günther, F.; Langer,M.;Muster, S.; Gouttevin, I.; Lange, S. Satellite-derived changes
    in the permafrost landscape of central Yakutia, 2000–2011: Wetting, drying, and fires. Glob. Planet. Chang. 2016,
    139, 116–127.
  • What about fire regimes in this region? Was this site affected recently by fires because this might be important for AL.
  • In a recent presentation at EGU, Georveski et al., showed that specific extreme climatic events might play an important role in permafrost degradation. Do you think that a specific extreme event can be decisive for AL deepening in Yakutia too?
  • Georgievski, G.; Hagemann, S.; Sein, D.; Drozdov, D.; Gravis, A.; Romanovsky, V.; Nicolsky, D.; Onaca, A.; Ardelean, F.; CheÈ›an, M. Climate extremes relevant for permafrost degradation. In EGU General Assembly 2020, 2020, 16115.
  • In the Discussion, you might also briefly refer to how climate change will impact the two types of environments analysed here: taiga and arable land. 

Author Response

Dear reviewer please see attached file.

the evolution of the AL was interpreted based on only 2 boreholes. Are there any other similar measurements in central Yakutia which are also in line with your findings? It might be risky to extrapolate your findings on a larger scale. A solution would be to compare your climatic data with ERA-Interim model (for example) and active layer with ESA CCI active layer thickness and to see whether the dependencies described by you are also available for the entire region. Otherwise, these findings have local importance in my opinion. Here are listed some studies which showed a good agreement between climate data operated by Roshydromet and ERA-Interim: Yes it is very interesting topic, but this study is very local.

  • You might also look at the land surface temperature evolution from MODIS data. In a recent paper (Boike et al. 2016) `the land surface temperatures showed a consistent warming trend, with an average increase of about 0.12 °C/year` in Central Yakutia. In the same paper snow cover is also assessed in this region, therefore it would be good to refer to this in the Discussion because several findings are similar to yours.  Added into the text
  • In a recent presentation at EGU, Georveski et al., showed that specific extreme climatic events might play an important role in permafrost degradation. Do you think that a specific extreme event can be decisive for AL deepening in Yakutia too? I could not find his presentation, just abstract
  •  

Round 2

Reviewer 2 Report

Thank you for addressing the changes I suggested. I believe that the authors have done a fine job during the revision of the first manuscript version, addressing most of the issues raised during the first refereeing round. The modifications add to the overall quality of the manuscript.

I added the link to the EGU presentation by Georgievski et al., 2020 below:

https://presentations.copernicus.org/EGU2020/EGU2020-16115_presentation.mp4

I think that you should add a paragraph on the role of extreme climatic events on ALD within the Discussions. 

Author Response

Dear reviewer,

The Georgievski is assed in the text.

Best regards

This manuscript is a resubmission of an earlier submission. The following is a list of the peer review reports and author responses from that submission.

Round 1

Reviewer 1 Report

This manuscript is an interesting attempt to quantify climate change in central Yakutia, concerning both the climate and the basic permafrost-related variables. Unfortunately, the quality of presentation appears mediocre, regarding both the English and the scientific value of this effort.

Basic data are interesting, but the reasoning on the relationship between contemporary climate and permafrost variables is weak and unconvincing. Trend analysis (Pearson/Mann-Kendall), change-point analysis, t-test or Welch test have sufficient power to support or disapprove most hypotheses put forward by the authors.

The manuscript structure also lacks consistency and logic. The Results and Discussion sections are not split into subsections, and the text flow is overall poor. Globally, this is an amateur effort to present what could be exceptionally interesting results, but lacking imagination, knowledge and compliance with scientific approach.

I advgise this manuscript be rejected straight away, and reworked from scratch to present scientifically viable results; besides, the English is extremely poor and requires extensive correction.

Some considerations are given in an attached manuscript file.

Author Response

Dear Reviewer this is information about correction according your suggestions.

It is a little difficult to explain corrections for all Your advices, so please sorry. The text is revised and all suggestions tried to be corrected.

English is improved as much us we can.

L2: The title is changed to thawing depth.

L11-23: The abstract is completely revised according to results.

L45: Added several new literatures.

L50: Added several new literatures.

L61: Figure is corrected with location

L162: Yes it is closest station to our site. This station is located on the edge of Yakutsk, so urban influence is not so high. The next closest station is located at more than 100 km away. On elevated another terrace of the Lena river. Yakutsk is shown on the map.

L162:

L205: Just no space.

L225:

L250:

Reviewer 2 Report

Comments for the Manuscript land-743157 “The impact of climate change on the thawing depth of zonal soils in Central Yakutia”

This manuscript elaborates with the already observed changes to the soil thawing depth of  Central Yakutia region. This is a very interesting work, based on valuable long term observations. This manuscript merit publication, however, there are some issues have to be resolved prior its publication.

Comments:

The introduction section has to be extended. The current literature review is poor and has to be extended, both in the context of climate change impacts on precipitation and temperature – as these two factors are presented to affect the thawing depth, as well as the potential feedbacks from soil temperature changes, as a controlling factor of biological processes. Furthermore, studies have shown that beyond the already observed changes, future projected warming is expected to reduce the Pergelic areas, hence this is worth discussing.

The paper urges for being checked for better English language use. Especially the abstract and the introduction sections.

Title: I would recommend to slightly modify the title into  The impact of “observed” climate change..

L16: maybe change or alteration sounds better than violation

L17: maybe destruction is an exaggeration. Changes in the thawing cycle may “destruct” the current status of equilibrium, which does not imply that another state of equilibrium would not appear and work as well.

L34: unstable equilibrium: this is a very contracting phrase. Something cannot be unstable and in equilibrium state at the same time. The sentence needs to be written more clearly.

L38: relevant to what?

L46: trends in increasing à “increasing trends in the mean annual.. “   Also if the 2.2C and 3C are the “trends”, then the time interval should also be mentioned, e.g. 2.2 / decade etc.

L161-169: I am wondering whether these abrupt changes in the temperature are also due to local climate factors, e.g. was there any abrupt change in the land use, or is there any other activity developed around that could affect the temperature? Or even a change in the measuring device?

Conclusions: Please change the list type within which the conclusions are started, with normal text discussion.

Also the paper would benefit by the comparison of the results to other studies elaboration with soil temperature regimes under a changing climate, e.g Climate-Induced Shifts in Global Soil Temperature Regimes 2016, and Climate change and the permafrost carbon feedback 2015.

Figure 2: In figure 2, authors identify a slightly positive trend due to an increase in October-November precipitation, however, is this change significant in a statistical manner, in order to be mentioned?

Figure 3A: Please change the color of the left y-axis to blue and the right to red to make it better discernable about which graph is what.

Author Response

Dear Reviewer this is information about correction according your suggestions.

It is a little difficult to explain corrections for all Your advices, so please sorry. The text is revised and all suggestions tried to be corrected.

The title was changed.

Introduction also was revised and corrected according your advice.

Within text was added some new literature, and some changes was done.

Conclusion is corrected.

Figures is changed and statistically analyzed.

English is improved as much us we can.

Round 2

Reviewer 1 Report

This is a second review of the manuscript by Desyatkin et al. The authors worked hard to enhance the paper, but multiple points still remain to be addressed before the manuscript could be accepted for publication. The main objective of the manuscript is clear - to relate regional climate changes with local active layer depth. This objective is still not attained. The authors

The soil description in Lines 95-180 is interesting, but is useless in the context of the paper – these data are never used to explain the variability in active layer depth change across the studied landscapes. What is then the goal of this description?

The climate data from Yakutsk meteo station, in their turn, are not presented in the ‘Materials and Methods’ section. What is the data source? Otherwise, I’ve tried to explore the openly available reanalysis data from PSL, and published in other sources, and these data does not correspond to what is presented in your analysis:

https://psl.noaa.gov/cgi-bin/data/timeseries/timeseries.pl?ntype=1&var=Air+Temperature&level=1000&lat1=62&lat2=63&lon1=129&lon2=130&iseas=1&mon1=0&mon2=11&iarea=1&typeout=2&Submit=Create+Timeseries

https://www.researchgate.net/publication/281931888_Climate_Change_Impact_on_Public_Health_in_the_Russian_Arctic/figures?lo=1

Statistical approach remains amateur. Statistical tests are used without consideration, and globally do not support the reasoning given in the paper. Figure 3 is a complete nonsence with these pretty Excel linear trends, especially essential nonsence in section D (snow depth) and followed by dubiously significant difference in corresponding means by t-tests in Table 3.

So, this manuscript does not present scientifically viable results, and fails to achieve its proper goal to provide evidences for a link between active layer depth and climate change in Yakutia. 

Example: during shift B, the relation between MAAT and ALD is positive linear, an increase in MAAT correponds to an increase in ALD; during shift C, this is no longer the case, higher MAAT does not result in higher ALD - this suggests a more complex relation between the two variables that you envisage.

My conclusion: reject, and encourage to resubmit after substantial revision.
